# Intersecting Pathways: Nonalcoholic Fatty Liver Disease and Psoriasis Duet—A Comprehensive Review

**DOI:** 10.3390/ijms25052660

**Published:** 2024-02-24

**Authors:** Daniel Octavian Costache, Horia Blejan, Damian Lucian Cojocaru, Georgiana Alexandra Ioniță, Marcela Poenaru, Maria Magdalena Constantin, Andrei Cătălin Costache, Constantin Căruntu, Daniel Vasile Balaban, Raluca Simona Costache

**Affiliations:** 1Discipline of Dermatology, Faculty of Medicine, Carol Davila University of Medicine and Pharmacy, 020021 Bucharest, Romania; daniel.costache@umfcd.ro (D.O.C.); maria.constantin@umfcd.ro (M.M.C.); 2Dermatology Department, Carol Davila Central Emergency Military University Hospital, 010825 Bucharest, Romania; horia.blejan@rez.umfcd.ro (H.B.); marcelapoenaru@yahoo.com (M.P.); 3Gastroenterology Department, Carol Davila Central Emergency Military University Hospital, 010825 Bucharest, Romania; damian-lucian.cojocaru@rez.umfcd.ro (D.L.C.); georgiana-alexandra.ionita@rez.umfcd.ro (G.A.I.); vasile.balaban@umfcd.ro (D.V.B.); raluca.costache@umfcd.ro (R.S.C.); 42nd Dermatology Department, Colentina Clinical Hospital, 020125 Bucharest, Romania; 5Faculty of Medicine, Carol Davila University of Medicine and Pharmacy, 020021 Bucharest, Romania; andrei-catalin.costache2023@stud.umfcd.ro; 6Discipline of Internal Medicine and Gastroenterology, Faculty of Medicine, Carol Davila University of Medicine and Pharmacy, 020021 Bucharest, Romania; 7Discipline of Physiology, Carol Davila University of Medicine and Pharmacy, 020021 Bucharest, Romania; 8Academy of Romanian Scientists, 050091 Bucharest, Romania

**Keywords:** psoriasis, NAFLD, nonalcoholic fatty liver disease, adipose tissue, pathogenesis pathways, biological therapy

## Abstract

Psoriasis is a chronic, immune-mediated, inflammatory disease that has a major impact on patients’ quality of life. Common psoriasis-associated comorbidities include cardiovascular diseases, psoriatic arthritis, inflammatory bowel syndromes, type-2 diabetes, and metabolic syndrome. Nonalcoholic fatty liver disease (NAFLD) is affecting a substantial portion of the population and is closely linked with psoriasis. The interplay involves low-grade chronic inflammation, insulin resistance, and genetic factors. The review presents the pathophysiological connections between psoriasis and nonalcoholic fatty liver disease, emphasizing the role of cytokines, adipokines, and inflammatory cascades. The “hepato-dermal axis” is introduced, highlighting how psoriatic inflammation potentiates hepatic inflammation and vice versa. According to the new guidelines, the preliminary examination for individuals with psoriasis should encompass evaluations of transaminase levels and ultrasound scans as part of the initial assessment for this cohort. Considering the interplay, recent guidelines recommend screening for NAFLD in moderate-to-severe psoriasis cases. Treatment implications arise, particularly with medications impacting liver function. Understanding the intricate relationship between psoriasis and NAFLD provides valuable insights into shared pathogenetic mechanisms. This knowledge has significant clinical implications, guiding screening practices, treatment decisions, and the development of future therapeutic approaches for these chronic conditions.

## 1. Introduction

Psoriasis is a chronic, immune-mediated, inflammatory disease that significantly impacts patients’ quality of life. The prevalence of psoriasis varies between 0.1% and 3% of the general population, with a minimum prevalence in East Asia and a maximum prevalence in the Scandinavian peninsula. Both genders are approximately equally affected, with the onset of the disease most common around the age of 20–30 years [1,2].

The pathogenesis of psoriasis is complex and not fully understood. However, thanks to the studies carried out in recent years, it has been shown that psoriasis is more than a skin disease, as the condition has a systemic impact on the body. Because psoriasis is associated with immunological dysfunction, patients suffer from chronic low-grade inflammation, which favors the development of multiple comorbidities [3]. These include cardiovascular diseases, psoriatic arthritis, inflammatory bowel syndromes, depression, neurological disorders, type-2 diabetes mellitus, and metabolic syndrome [4,5].

One association that has come to the attention in recent years is that between psoriasis and nonalcoholic fatty liver disease (NAFLD). NAFLD covers a broad spectrum of manifestations, ranging from a simple accumulation of fat in the liver to steatohepatitis (NASH), which involves inflammation of the liver parenchyma, the ballooning of hepatocytes, and liver fibrosis, which can progress to cirrhosis and hepatocellular carcinoma. It is important to mention that, for the diagnosis of NAFLD, any cause of liver injury must be excluded [6]. It is currently estimated that about one-third of the general population suffers from NAFLD, and it is the main cause of abnormal liver enzyme levels in the Western population [7].

The link between psoriasis and NAFLD is not fully elucidated, but, considering that patients with psoriasis have low-grade chronic inflammation and increased insulin resistance, which are key elements for the development of NAFLD, the connection between the two conditions seems plausible. This association is relevant to everyday practice. First of all, once the diagnosis of NAFLD is suspected, most frequently due to changes in liver enzyme values, the patient must undergo a series of investigations for the correct staging of the disease and assessment of the associated risk. Secondly, the association between psoriasis and NAFLD has implications for the choice of systemic antipsoriatic therapy. It is known that some antipsoriatic medications commonly used in daily practice cause liver injury and may, thus, enhance the negative effect already induced by NAFLD, leading to liver-disease progression [8,9].

This review aims to better understand the common pathophysiological mechanisms between psoriasis and NAFLD, as well as to assess, based on existing data from the literature, whether there is a causative relationship between the two conditions and to evaluate the therapeutic implications that this possible association has for the treatment of patients with psoriasis.

## 2. Diagnosis of NAFLD

In light of the latest available evidence, the new academic guidelines propose that individuals exhibiting moderate-to-severe psoriasis should undergo screening for NAFLD when deciding on a psoriasis treatment. According to these guidelines, the initial assessment for this patient population should include transaminase-level assessments and ultrasound scans into the initial evaluation for this group, along with proposing a continuous care plan that involves referring patients to a hepatologist if there are indications of potential liver disease [10].

The diagnosis of NAFLD requires a comprehensive approach to exclude potential causes of steatosis. After ruling out common contributors, such as daily alcohol consumption exceeding recommended limits and other steatogenic factors (including certain medications and parenteral nutrition), the exclusion of viral hepatitis, iron overload, autoimmune liver disease, Wilson’s disease, inborn metabolic disorders, pregnancy in cases of high clinical suspicion are crucial steps in definitively identifying NAFLD [11].

Ultrasound is acknowledged as the primary method for assessing steatosis in individuals suspected of having NAFLD, and it is recommended as a screening test by the existing guidelines [12]. Usually, a fatty liver manifests as “hyperechoic” in ultrasound imaging due to enhanced acoustic interfaces in hepatic lipid accumulation, an amplified difference in echo amplitude between the liver and kidney, and the absence of echoes from the walls of the portal system [13]. Analyzing proton signals to differentiate between fat and water signal components based on their resonance frequency, magnetic resonance spectroscopy (MRS) stands out as a highly precise noninvasive technique for assessing liver fat content. Nevertheless, its broad application faces notable constraints, such as the necessity for specialized proficiency in executing and interpreting MRS [14].

Advancements in magnetic resonance technology have played a crucial role in precisely and noninvasively quantifying steatosis and fibrosis. MRS, in particular, has been instrumental in the detection of fat by quantifying the proton density fat fraction (PDFF), a standardized measure of liver steatosis [15]. The utilization of the controlled attenuation parameter (CAP) is increasingly prevalent as an ultrasound-based tool for assessing liver fat. This parameter can be acquired concurrently with a liver stiffness measurement (LSM) using vibration-controlled transient elastography (VCTE), commercially known as FibroScan, used for the measurement of fibrosis and steatosis [16].

Acoustic radiation force impulse (ARFI) represents an innovative and dynamic internal method, integrating information on tissue elasticity into the standard grayscale imagery produced by widely-used ultrasound (US) scanners. Notably, ARFI demonstrates a markedly reduced incidence of unsuccessful measurements compared to transient elastography (TE), indicating its enhanced reliability in clinical applications [17].

In NAFLD patients, liver biopsy remains the gold standard for evaluating steatosis, steatohepatitis, and fibrosis and is critical in determining clinical outcomes when there is diagnostic uncertainty. However, the effectiveness of liver biopsy is hindered by recognized limitations, including costs, potential sampling errors, inter-/intraobserver variability, and risks associated with the procedure [18]. The histological examination of NAFLD indicates the presence of centrilobular steatosis in at least 5% of liver tissue. On the contrary, NASH is characterized by steatosis exceeding 5%, along with the presence of an inflammatory infiltrate and a ballooning of hepatocytes [19].

Serum enzyme levels, specifically alanine aminotransferase (ALT) and aspartate aminotransferase (AST), may exhibit mild elevation, with a higher frequency observed in NASH compared to NAFLD. Alkaline phosphatase levels may experience a minor increase and levels of gamma-glutamyl transferase (GGT) are regularly raised [20].

FibroMax encompasses FibroTest, ActiTest, and SteatoTest. The differences between them are that FibroTest is calculated by considering the concentrations of α2-macroglobulin, haptoglobin, apolipoprotein A1, total bilirubin, and GGT, with adjustments made for age and sex. ActiTest, on the other hand, includes ALT in addition to these parameters. Furthermore, SteatoTest incorporates total cholesterol, fasting glucose, triglycerides, height, and weight into its calculation. The studies suggest that Fibromax could serve as a viable alternative to liver biopsy [21].

The Fatty Liver Index (FLI) has been proven effective in diagnosing NAFLD and is composed of BMI, waist circumference, and serum levels of triglycerides and GGT. The assessment of liver fat content through the NAFLD Liver Fat Score (NLFS) has proven to be accurate in diagnosing NAFLD and is computed considering factors such as metabolic syndrome, type-2 diabetes, fasting serum insulin, and AAR [22].

Inflammatory markers, including ferritin and high-sensitivity C-reactive protein (CRP), have been linked to NASH. Analyzing cytokeratin 18 (marker indicating hepatocyte caspase-3 activation within the apoptosis pathway) fragments in serum demonstrates the capability to anticipate histologically confirmed NASH and assess the severity of the disease in a significant and diverse population of individuals with NAFLD [11,23].

Two of the most used scores to diagnose fibrosis are FIB-4 (AST, ALT, age, and platelet count) and NFS (age, BMI, diabetes mellitus/IFG, albumin, platelet count, and De Ritis ratio). Still, they can also identify steatosis with a positive predictive value of 65% and 82–90%, respectively [23].

## 3. Common Pathways in the Pathogenesis of Psoriasis and NAFLD

Psoriatic skin eruptions are the hyperproliferation of a layer of the skin, the epidermis, due to abnormal differentiation of keratinocytes, leading to parakeratosis and acanthosis. Keratinocytes (KC) are stimulated by an inflammatory infiltrate consisting of T lymphocytes, dendritic cells, macrophages, and neutrophils. Various inflammatory cascades further develop, ultimately resulting in a state of chronic inflammation [23].

In recent years, adipose tissue and the gastrointestinal tract have contributed as critical factors in the inflammation and fibrosis underlying NAFLD. Even though inflammation plays such an important role in the pathogenesis of NAFLD, there are no large randomized controlled trials specifically targeting inflammatory pathways [24].

An altered liver function involving, for example, hepatic insulin resistance and hepatic lipotoxicity are considered another hallmark of NAFLD [25]. Interactions between the liver, adipose tissue, and gut are bidirectional, as established by functional experiments using transgenic mouse models, which is exemplified by a multitude of studies [26]. Collectively, evidence over the past decade has corroborated Tilg and Moschen’s hypothesis of multiple parallel hits reported in 2010 into a concept that builds on mechanistic knowledge gained in animal models and descriptive clinical studies [24].

Adipose tissue acts as an endocrine organ, resulting in the formation of cytokines associated with both psoriasis and NAFLD (adipocytokines, adipokines (leptin and resistin) TNF-α, and IL-6) [27]. Proinflammatory adipokines (TNF-α, IL-1, IL-6, and leptin) play a role in keratinocyte proliferation, and IL-1 is involved in molecular adhesion in the pathophysiology of psoriasis. In contrast, their role in NAFLD is based on increased liver fibrogenesis, contributing to increased insulin resistance. IL-1 also plays an important role in increasing the activation of MAP and ERG pathways [10,28] (Figure 1).

Another study identifies that patients with NAFLD and psoriasis have elevated CRP and IL-6 values, but low adiponectin values [10]. Adiponectin has an important role in metabolism; it decreases hepatic and systemic insulin resistance and attenuates inflammation and fibrosis in the liver. Therefore, at lower levels of adiponectin, the liver will have decreased protection from the aggression of proinflammatory factors [10,28].

Proinflammatory hepatokines (FGF 21, Fetuin-A, CRP, IL-6, and TNF-α) are responsible for increasing liver fibrogenesis and increasing susceptibility to liver-cell injury [20,23]. These mediators allow a cross-association between adipose tissue and the liver, which responds by producing hepatokines (fibroblast growth factor 21, Fetuin-A, CRP, TNF, and IL-6) [10]. In patients with psoriasis, there is an imbalance between the levels of pro- and anti-inflammatory adipokines (TNF-α, IL-6, leptin, resistin, and vifastin which increases, and adiponectin which decreases).

Insulin resistance can be caused by the overproduction of proinflammatory cytokines in patients with psoriasis, resulting in the accumulation of triglycerides in the hepatocytes. This leads to steatosis, which makes the liver susceptible to further inflammatory aggression, leading to mitochondrial dysfunction and oxidative stress. All of these inflammatory processes disrupt hepatocyte apoptosis by activating TGF- and hepatic stellate cells, generating a vicious loop in which steatosis leads to greater insulin resistance via the inflammatory cascades it creates [23].

As a result, an imbalance between the synthesis and disposal of triglycerides may emerge, leading to progressive steatosis and NASH, while another imbalance develops between hepatocyte death and regeneration. The inflammatory processes associated with psoriasis, as well as the altered TNF-adiponectin ratio, accelerate the course of liver disease [27].

Some articles show that psoriasis is associated with NAFLD independently of metabolic syndrome and its components, due to the genetic complexity of NAFLD [29]. The I148M polymorphism within phospholipase plays an important role in the susceptibility to NAFLD, and the TM6SF2 polymorphism increases the risk of liver fibrosis [30,31,32,33]. Vitamin D has a major influence in both conditions and its deficiency potentiates the activation of Toll-Like receptors that increase liver inflammation. Vitamin D also contributes to the regulation of the skin’s immune system [34].

Common risk factors contributing to the pathogenesis and association of psoriasis and NAFLD are persistent inflammation, imbalance of proinflammatory and anti-inflammatory cytokines, genetic factors through polymorphisms, and crossover. The presence of factors characterizing the metabolic syndrome (obesity, low insulin resistance, and dyslipidemia/hyperlipidemia with increased triglycerides in the liver cell) is also implicated in the development of these conditions [23]. There is a reciprocal action between the psoriatic inflammatory process and NAFLD based on the role of cytokines. The pathway from steatosis to steatohepatitis is facilitated by TNF-α and IL-6, and TGF-β increases the risk of liver fibrosis [35].

IL-17 contributes to the pathogenesis of both psoriasis and NAFLD, as this cytokine is secreted by Th17 cells (which can be detected in adipose tissue). IL-17, among other roles, is involved in the regulation of glucose metabolism and adipogenesis [10,36]. When inflammation is present in the liver, CD4+T (Th17) and CD8+T (Tc17) cells produce interleukin IL-17, which can also be produced by macrophages, natural killer cells, neutrophils, and Ty gamma cells. Increased IL-17 levels could be correlated with faster progression from MAFLD (metabolic-associated fatty liver) to NASH or even hepatocellular carcinoma [10,36].

Some studies show that inhibiting IL-17 in patients suffering from psoriasis can help them with decreasing liver aggression and, thus, improve metabolism. Being a key factor, IL-17, along with other inflammatory cytokines, may underlie new effective therapies for both psoriasis and NAFLD [36]. Obesity negatively promotes liver disease by increasing Th17 cell levels and IL-17 production. Increased levels of IL-17 enable activation of the IL-17 receptor in hepatocytes, Kupffer cells, and stellate cells, resulting in increased production of proinflammatory cells and reactive oxygen species [10,35,37].

IL-17 is not necessarily a powerful inflammatory factor in and of itself; rather, it functions through its ability to stimulate immune-cell recruitment and chemokine and receptor potentiation, as well as by functioning synergistically with other cytokines, initiating a proinflammatory cascade with them. Its basic role is to protect against pathogens, but excessive activation through various pathways causes it to switch to a proinflammatory state [10,23]. An important role in the pathophysiology of psoriatic disease is played by the IL-17–Th17 axis through IL-1 dendritic cells that potentiate and differentiate Th17 lymphocytes, resulting in increased production of IL-17 and other cytokines that have a direct effect on epidermal keratinocytes [10].

The main link between psoriasis and NAFLD seems to be low-grade inflammation. Psoriasis promotes systemic inflammation which, in turn, promotes inflammation in adipose tissue, leading to the release of adipokines (resistin, leptin, and visfatin) and proinflammatory cytokines (IL-6 and TNF-α), which are directly linked to increased insulin resistance. At the same time as this process described, there is also a decrease in anti-inflammatory adipokines (adiponectin). This imbalance between pro- and anti-inflammatory cytokines causes a proinflammatory feedback loop between skin, adipose tissue, and liver. This cooperation of the three organs configures the so-called hepato-dermal axis, which means that psoriatic inflammation potentiates hepatic inflammation and vice versa [10,23].

A recent study has put forward the idea that bacterial translocation from the skin and gut of psoriasis patients is associated with NAFLD and with the potentiation of the immune response by increasing intestinal permeability, which may be mediated by the microbiome [38]. Another study has shown that IL-12 also helps to accumulate and activate lymphoid cells in adipose tissue, thus causing local inflammation. It was found that there is a correlation between IL-12 levels and an increased BARDI score (predictive score for fibrosis in NAFLD patients), which is based on BMI, AST/ALT, and the presence of diabetes [39].

Another factor contributing to increased inflammation is inflamed visceral adipose tissue, which perpetuates chronic inflammation and increased release of nonesterified fatty acids, which in turn contribute to the development of psoriasis. No doubt visceral adipose tissue plays a key role in the development of insulin resistance, inflammation, and NAFLD, possibly through the secretion of multiple factors, such as the increased release of nonesterified fatty acids, increased production of various hormones, and proinflammatory adipocytokines (including TNF-α, IL-6, leptin, visfatin, and resistin) and decreased production of adiponectin. In the presence of obesity and insulin resistance, there is an increased influx of nonesterified fatty acids into the liver. There is now substantial evidence that nonesterified fatty acids play a crucial role in directly promoting liver injury by increasing intrahepatic oxidative stress and activating the inflammatory pathways [40].

As a result of all the inflammatory processes it produces, IL-17 remains a key factor in the low-grade inflammation that underlies the immunological association between psoriasis and NAFLD [10].

These two pathologies, psoriasis and NAFLD, appear to address the same pathogenetic mechanism in which IL-17 plays a key role. Metabolic syndrome is also based on the same inflammatory processes. Understanding these overlapping inflammatory cascades underlying the pathogenesis of psoriasis/NAFLD may help to develop future therapeutic approaches and common strategies [10,23].

## 4. Therapeutic Implications

Given the fact that psoriasis is a chronic condition and there is currently no curative treatment, patients are forced to take medication for the long term. In general, the therapeutic option is chosen according to the severity of the disease, possible comorbidities present, and the therapeutic option agreed upon by the patient. Fortunately, there are several therapeutic options for these patients, and this allows the dermatologist to choose an individualized therapy according to the patient’s needs [41,42]. Table 1 shows the main therapeutic classes available for the treatment of psoriasis depending on the severity and individual needs of each patient.

Considering that psoriasis and NAFLD share a common pathophysiological mechanism based on low-grade chronic inflammation, certain antipsoriatic therapeutic options that reduce inflammation may have a beneficial effect on both conditions [43].

The following section will discuss the main therapeutic options for patients with psoriasis, as well as the possible risks and benefits that patients with psoriasis and NAFLD are exposed to following therapies.

Topical corticosteroid therapy and phototherapy are good therapeutic options for patients with mild forms of psoriasis. These therapies do not appear to alter liver enzyme parameters or increase the risk of metabolic syndrome and are, therefore, safe and without any negative impact for patients with psoriasis and NAFLD [44].

Methotrexate was and is one of the most prescribed systemic drugs for the treatment of moderate and severe forms of psoriasis. It has both an antiproliferative action and an immunosuppressive effect by decreasing inflammation, which, in theory, should benefit both psoriasis and NAFLD [45]. However, methotrexate stimulates the synthesis of pro-fibrotic factors in hepatocytes, which can lead to liver fibrosis [46]. Another interesting finding in the literature is that methotrexate-induced liver injury occurs more frequently in patients with psoriasis than in those with rheumatoid arthritis, even though both conditions are based on low-grade chronic inflammation. However, this finding can be partly explained by the fact that patients with psoriasis have a higher BMI than those with rheumatoid arthritis and a higher prevalence of metabolic syndrome [47]. In a study by Rosenberg et al., patients with psoriasis associated with type-2 diabetes and obesity had a much higher risk of developing liver fibrosis when treated with methotrexate [48]. In a comparative study looking at the safety of methotrexate vs. secukinumab in patients with psoriasis and metabolic syndrome, it was found that patients given 15 mg per week of methotrexate had increased liver enzyme values at 6- and 12-month follow-ups, with some patients having values up to three-times higher than normal. In contrast, patients treated with secukinumab showed no change in liver enzyme values, thus having a neutral effect [49].

Although the toxic effect of cyclosporine on the kidney is well known and documented, the possible negative repercussions on the liver should not be neglected. Depending on the dose and duration of cyclosporine administration, patients may experience an increase in serum bilirubin and liver enzymes, which may culminate in liver injury in certain circumstances [4]. There is conflicting information on the hepatotoxic effect of cyclosporine. It is hypothesized that cyclophilin inhibition may have a hepatoprotective effect in patients with cirrhosis or established liver fibrosis [50]. In another study by Gisondi et al., patients treated with cyclosporine showed low liver enzyme values but elevated total cholesterol and triglyceride values at 16-week follow-up [51,52]. However, cyclosporine-treated patients are at increased risk of developing type-2 diabetes mellitus, which is a risk factor for metabolic syndrome and, thus, indirectly a risk factor for NAFLD. This phenomenon can be partly explained by the diabetogenic effect of cyclosporine, which inhibits insulin secretion from the pancreas [52].

Acitretin, a vitamin A derivative, and a drug commonly used for patients with moderate to severe psoriasis, can have negative effects on the liver. Studies show that up to a quarter of patients treated with acitretin may experience transient changes in liver enzymes, hypertriglyceridemia, and hyperlipidemia. However, a small number of patients also develop liver injuries [53]. This observation is supported by a cross-sectional study in which patients treated with acitretin or methotrexate did not show an increased risk of developing steatohepatitis. In general, patients who may benefit from Acitretin therapy are those who do not have NAFLD or those who do not have risk factors for NAFLD [54]. In a small clinical trial study, acitretin interestingly showed a decrease in insulin resistance. This could be very beneficial for medical practice, as insulin resistance is the cornerstone of both NAFLD and psoriasis and could have a beneficial effect on both conditions. However, more studies with larger groups of patients are needed to confirm the above-mentioned effect [55].

Apremilast is a phosphodiesterase 4 inhibitor that can also be used in patients with moderate to severe forms of psoriasis. In randomized clinical trials, this drug did not appear to have any negative effects on the liver or liver enzymes. Moreover, this drug seems to improve the parameters of carbohydrate and lipid metabolism, and in obese patients, it can lead to weight loss of up to 5% of the patient’s weight. This effect has been observed in about 20% of psoriasis patients treated with apremilast. Adverse effects observed in patients treated with apremilast include insomnia, depression, suicidal ideation, and diarrhea [56,57,58].

Biological therapies are currently the newest class of drugs that dermatologists can use to treat severe psoriasis that does not respond to conventional systemic therapies. There are currently four categories of US Food and Drug Administration (FDA)-approved biological drugs for use in psoriasis: tumor necrosis factor inhibitors (TNFi) (etanercept, infliximab, adalimumab, and certolizumab); interleukin IL-12/23 antagonists (ustekinumab); IL-17A inhibitors (secukinumab and ixekizumab); and anti-IL-23 agents (tildrakizumab, risankizumab, and guselkumab) [59]. Biological therapies are considered to have a better safety profile than conventional systemic medications, and the choice of the therapeutic class should be individualized and take into account any associated comorbidities [60].

TNF-α plays an essential role in the pathogenesis of psoriasis as well as NAFLD. In theory, TNF-α inhibition should benefit patients with psoriasis and NAFLD by decreasing insulin resistance, which is central to both conditions [6]. This result is invariably demonstrated in studies conducted to date. Nevertheless, there are optimistic premises suggesting that etanercept administration could improve insulin resistance in these patients and, thus, lower their risk of developing type-2 diabetes. Etanercept also seems to inhibit the secretion of proinflammatory adipocytokines and modulate the levels of A1 and B apolipoprotein [61,62,63].

A prospective study, with follow-up periods at 8 and 16 weeks, conducted in Italy followed the effects of various systemic therapies on liver enzyme values and lipid profiles in patients diagnosed with psoriasis. In this study, it was observed that patients treated with etanercept showed slightly increased triglyceride values at both 8- and 16-week follow-ups. This is in agreement with information in the scientific literature confirming a slight increase in triglyceride and total cholesterol values among patients treated with TNF-α inhibitors. At the 16-week follow-up, it was observed that patients treated with infliximab showed increased liver enzyme values, with some patients even doubling their values compared to the reference values. Among the available biological therapies in the TNF-α inhibitor class, the most evidence of hepatotoxicity points to infliximab. However, liver enzyme values return to normal upon discontinuation of infliximab, so a therapeutic switch to another TNF-α inhibitor seems to promise good results [51]. Autoimmune hepatitis following the administration of TNF-α inhibitors is a very rare but noteworthy phenomenon. Most cases have occurred following infliximab administration and have led, among other things, to increased liver enzyme levels and a decrease in HDL cholesterol. Most of the cases recovered when infliximab was stopped, and a short course of systemic corticosteroids was administered [64].

A further interesting finding in patients treated with TNF-α inhibitors is weight gain, which was not observed in patients treated with IL-12/23 blockers. Weight gain seems to be directly correlated with the duration of TNF-α inhibitor administration, so patients treated for long periods with TNF-α blockers showed more significant weight gain [40]. In another study conducted over a 5-year period in which 29 patients with psoriasis concomitantly associated with liver injury (including liver C virus infection) were treated with adalimumab, it was observed that no patient showed progression of liver injury [65].

IL-17 is an essential cytokine in the pathogenesis of psoriasis and NAFLD, so its inhibition should have beneficial effects on both pathologies. In mouse models, the above-mentioned theory is supported by the fact that low levels of IL-17A lead to a decrease in insulin resistance, thus being a protective factor for the development of NAFLD. Phase III clinical trials of ixekizumab and secukinumab showed an adequate safety profile with no adverse hepatic effects [20,23,66]. In an exploratory post hoc analysis, the effects of systemic antipsoriatic therapies on liver enzymes and various parameters characteristic of the metabolic syndrome were examined. The study compared secukinumab, etanercept, and placebo. Patients treated with secukinumab showed a slight weight loss at 52 weeks of follow-up; in comparison, patients treated with etanercept showed a weight gain, which is consistent with information available in the literature. Triglyceride, total cholesterol, LDL, and HDL values remained unchanged in patients treated with IL-17 blockers. The same trend could be observed when liver parameters were evaluated. Secukinumab did not influence liver parameters in any way. In contrast, patients treated with etanercept showed an increase in liver enzymes that could be detected as early as week 16 of follow-up [67].

IL-12/23 inhibitors are another therapeutic class used for the treatment of patients with severe psoriasis. IL-23 is the linking cytokine between innate and adaptive immunity and plays an essential role in maintaining inflammation, while IL-12 supports the production of cytokines essential in the pathogenesis of psoriasis, such as TNF and interferon-γ [68]. Ustekinumab is the main representative of the class. A study conducted in Spain on a group of 44 patients treated with ustekinumab followed the effect of biological therapy on liver parameters. Of the total number of patients, only six showed a slight increase in liver enzymes without jeopardizing the patient’s health or the therapeutic decision to change medication. The study also showed that ustekinumab is a feasible option for patients with pre-existing liver disease or who have developed liver injury during other biological therapies [69]. In a comparative study, ustekinumab was more effective than etanercept in patients with moderately severe forms of psoriasis. This was supported by lower PASI scores and the physician’s global assessment 12 weeks after starting therapy [70]. An interesting aspect worth mentioning is that ustekinumab seems to be less effective in patients with a high BMI. In a study conducted in Japan on a group of 111 patients, the number of patients with a reduction in the PASI score of 90 or 100 was lower among patients with a BMI greater than 25. BMI is known to be directly correlated with psoriasis severity. This may have important implications for the choice of therapeutic class, as it is known that patients with psoriasis are heavier than the general population [71].

For the new molecules in the IL-23 inhibitor class (guselkumab, risankizumab, and tildrakizumab), the data on the hepatic safety profile are few but optimistic, as these molecules do not appear to produce liver injury. More studies need to be done to assess its risk [4,72].

Considering the information presented above, biologic therapy has a good hepatic safety profile, offering the physician a safe alternative to conventional systemic therapy. Moreover, biologic therapy has been shown to have a hepatoprotective role even in psoriasis patients with pre-existing liver injury, especially the newer therapeutic classes (IL-17 inhibitors and IL-12/23 inhibitors). The choice of therapeutic class should be made by an experienced physician who considers all possible parameters that may influence liver health. So far, the new therapeutic classes have shown hepatoprotective benefits for patients with psoriasis and nonalcoholic fatty liver disease.

## 5. Correlation between Psoriasis and NAFLD

The mechanism by which psoriasis associates multiple comorbidities is not fully understood, but as mentioned above, low-grade chronic inflammation plays a central role [73]. This was demonstrated in the Taiwanese study which showed that the prevalence of NAFLD is higher among patients with early-childhood-onset psoriasis. Patients with early-onset disease have a much more severe systemic inflammatory response which, in turn, can aggravate psoriasis [74].

The occurrence of these conditions seems to be influenced by an individual genetic susceptibility as well as by potential risk factors that the patient may associate with. NAFLD can be considered an extension of the metabolic syndrome in the liver [20]. In a study of patients over the age of 55, it was found that 46% of patients with psoriasis met the diagnostic criteria for NAFLD, compared to only 33% of patients in the reference group [29]. Potential risk factors for developing NAFLD were found to be male gender, obesity, hyperglycemia, and hypertension [75]. In a cohort of 218 patients diagnosed with psoriasis, 113 also had NAFLD. The prevalence of patients who had both psoriasis and NAFLD was significantly skewed in preference to male patients (74.3%) [76]. The risk of developing NAFLD among psoriasis patients appears to be increased regardless of the origin of the study group (Asia vs. Europe vs. North America). However, the prevalence of the disease varies quite a lot from study to study. This can be explained primarily by the different diagnostic methods used to confirm the disease. Secondly, it is well known that each ethnic group has a different prevalence of each disease. In the case of NAFLD, a higher incidence could be observed among Hispanic patients and a much lower incidence among African Americans [77,78]. In numerous studies, including the 2014 American study, it has been shown that the highest prevalence of NAFLD is among Hispanic men [79]. Also, in a study conducted in Spain that monitored the prevalence of NAFLD among patients with psoriasis, it was found that the patients most likely to develop NAFLD were men with a high BMI and a large waist circumference. In this study, it was not possible to demonstrate a correlation between a patient’s age and the risk of developing NAFLD. Only 14% of the patients studied had liver fibrosis. Another interesting fact observed was that patients treated with methotrexate did not show an increased risk of developing hepatic steatosis. In line with the above-mentioned statement, Miele et al. did not observe an increased risk of NAFLD in patients treated with acitretin. However, even in the absence of these risk factors, patients with psoriasis have an independent risk of developing NAFLD [54,75].

There are numerous studies in the literature on the possible correlation between psoriasis severity and NAFLD progression. One study that supports this hypothesis is that of Gisondi et al. which claims that patients who associate both diseases have a higher prevalence of metabolic syndrome. More than this, the PASI score among these patients was higher than those with psoriasis alone [57]. Likewise, in a meta-analysis of 15 observational cross-sectional studies, it was observed that patients with psoriasis have a 2-fold higher risk of developing nonalcoholic fatty liver disease compared to the general population. Also, in this study, an interesting finding was that the severity of psoriasis appeared to be directly correlated with the risk of developing NAFLD, an aspect on which there is no consensus in the literature [80]. In antithesis to the studies mentioned above, Candia et al. also reported a 2.1-fold increased risk of developing NAFLD, but in this study, the risk was not correlated with psoriasis severity [77].

In another study from Iran, there was a notably elevated prevalence of nonalcoholic fatty liver disease (NAFLD) in individuals with psoriasis compared to their counterparts without the condition (65.6% vs. 35%). Furthermore, psoriasis emerged as a standalone predictor of the grade of NAFLD, emphasizing its independent association with the severity of liver involvement [78]. The same observation was made by Phan et al. in their meta-analysis of nine cross-sectional studies [75].

Even though the PASI score is the most frequently used tool for assessing psoriasis severity, some studies rely on the BSA in analyzing the correlation between psoriasis severity and NAFLD. In an Indonesian study from 2020, the prevalence of NAFLD in psoriasis patients was approximately 77.8%. An interesting fact to note is that, in this study, a more significant correlation was shown between psoriasis severity based on BSA (body surface area affected by psoriasis) than based on PASI score [80]. In another study, a weak correlation was described between psoriasis severity based on PASI and NAFLD based on CAP, but there was a significant association between BSA/CAP. The strength of this study is that it is the first in the world to use CAP, which has higher specificity than ultrasonography [81,82].

Approximately one-third of the population in developed countries is diagnosed with NAFLD [40]. Even so, the prevalence of the disease may be higher. NAFLD has no specific signs and symptoms, and laboratory tests can be misleading in that up to two-thirds of patients with NAFLD have liver markers within normal limits. In general, ALT and AST values are not much higher than the reference values (on average, 2–3 times the threshold value), with an ALT–AST ratio above one [65]. An interesting aspect observed in a study conducted in Romania in 2020 was that patients who only have NAFLD have lower AST/ALT and fibrosis scores than patients who have concurrent psoriasis [83]. This demonstrates that the association between NAFLD and psoriasis leads to an unfavorable prognosis in the evolution of liver disease. Likewise, in an Italian cross-sectional study from 2015, the increased prevalence of NAFLD (44%) was demonstrated in the case of psoriatic patients. An interesting fact is that the risk of developing liver fibrosis is higher among patients suffering from psoriasis than among the control group according to the NAFLD-Fibrosis Score [57]. One less studied serum marker in the context of the association between psoriasis and NAFLD, but still important, is uric acid in peripheral blood. Studies in East Asia and the USA have shown that serum uric acid levels appear to be directly correlated with the risk of developing NAFLD. Moreover, this risk appears to be independent of other known risk factors for NAFLD [84].

One of the comorbidities frequently encountered in patients with psoriasis is psoriatic arthritis. In an American study from 2014, which includes a cohort of 103 patients suffering from psoriasis or psoriatic arthritis, 46.6% also suffer from NAFLD. Serum markers, such as transaminases, lipid profile, glucose, insulin, HA1C, and ferritin, were considerably increased in the group presenting both pathologies. Also, patients suffering from NAFLD had a much higher PASI score than those without NAFLD [79,85]. On the other hand, a 2019 study aimed to compare the prevalence of NAFLD in patients with psoriasis and those with psoriatic arthritis provides the fact that the occurrence of psoriatic arthritis does not increase the risk of metabolic complications as well as NAFLD. This can be explained by the fact that patients with psoriatic arthritis are generally treated with systemic biologic therapy, which has a superior safety profile to classical systemic therapy [9,60].

A curious finding is that patients under biological therapy for psoriasis have a lower cardiovascular risk [86]. Another interesting aspect of biological therapy was observed in a study conducted in Italy; out of 112 patients suffering from psoriasis, 84 had concurrent NAFLD. Among the patients who followed for 12 months of treatment with anti-TNF-α antibodies, 75% of those who followed treatment with Etanercept also developed NAFLD. It is important to mention the fact that no patient underwent treatment with corticosteroids. As in other studies, most patients also had metabolic syndrome, dyslipidemia, and obesity [87].

A 2016 Turkish study conducted with 518 psoriatic patients aimed to demonstrate the association between psoriasis and liver damage. Among all the therapies analyzed in this study, it was shown that 57% of the modified liver tests were in the case of patients treated with methotrexate [88]. Another study in Spain that followed the same objective showed that patients treated with cyclosporine or methotrexate had an increased rate of liver injury [89].

## 6. Conclusions

NAFLD is a disease with an increasing prevalence. The association with psoriasis has an unfavorable prognosis for both pathologies because both conditions increase the risk of reciprocal severity. This is closely related to local and systemic inflammatory reactions, plus other risk factors such as metabolic syndrome or genetic polymorphism that potentiate this cycle of inflammation. More specifically, the cooperation between skin, adipose tissue, and liver forms a hepato-dermal axis that is the basis of the constant potentiation of inflammatory cascades. These facts mean that psoriatic patients should be routinely screened for NAFLD, and referral to a hepatologist for additional evaluation should be considered. A variety of noninvasive imaging methods are available to diagnose NAFLD, from ultrasonography as a primary screening investigation to MRS-PDFF or elastography. However, liver biopsy remains the gold standard for assessing steatohepatitis and liver fibrosis. The addition of biological therapies has been a ground-breaking change in the treatment of psoriasis, and the safety profile of these medications appears to be superior to classical systemic medications. Moreover, some drugs appear to have a beneficial effect on both psoriasis and NAFLD. Thus, it is important to use a tailored approach regarding the medication chosen according to the comorbidities and risk factors associated with each patient, which can have a double beneficial effect for the patient.

More prospective studies with larger numbers of participants are needed to show the true impact of this association between psoriasis and NAFLD. Further investigation is needed to determine whether NAFLD is only an epiphenomenon of coexisting features of metabolic syndrome or is an independent risk factor for the development and progression of psoriasis. Considering the strong association between NAFLD and psoriasis, we believe that healthcare practitioners following psoriatic patients should be alert to this potentially progressive liver disease that is commonly seen among psoriatic patients.

## Figures and Tables

**Figure 1 ijms-25-02660-f001:**
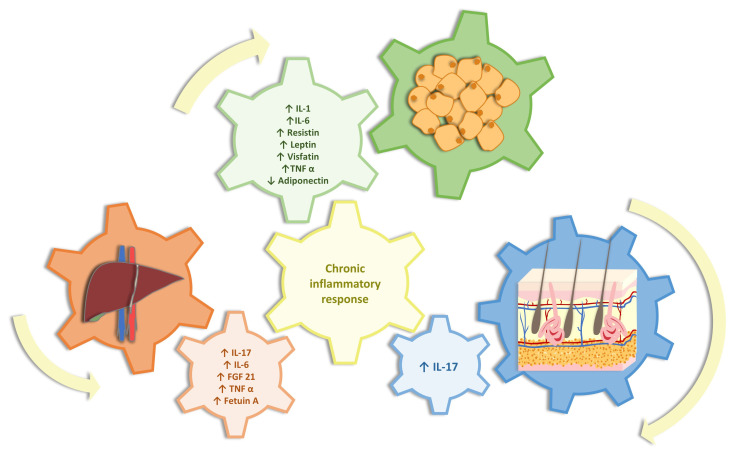
The role of the hepato-dermal-adipose axis in the duet between nonalcoholic fatty liver disease and psoriasis (IL: interleukin; TNF α: tumor necrosis factor-alpha; and FGF 21: fibroblast growth factor 21).

**Table 1 ijms-25-02660-t001:** Therapeutic options for psoriasis treatment.

Topical Treatment	Phototherapy	Systemic Treatment	
		Conventional	Biologic
Corticosteroids	UVB-NB	Methotrexate	TNF inhibitors
Calcineurin inhibitors	PUVA	Cyclosporine	IL 12/23 inhibitors
Vitamin D analogues	rePUVA	Acitretin	IL 17 inhibitors
Retinoids		Apremilast	IL 23 inhibitors
Coal tar			
Anthralin			
Emollients and Moisturizers			
Keratolytic agents			

## Data Availability

No new data were created.

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
