# Peer review of "Intersecting Pathways: Nonalcoholic Fatty Liver Disease and Psoriasis Duet—A Comprehensive Review"

_ijms, 2024, doi:10.3390/ijms25052660_

Round 1
Reviewer 1 Report
Comments and Suggestions for Authors
The authors wrote a comprehensive review on the topic of non-alcoholic fatty liver disease (NAFLD) and psoriasis. They analyze common pathophysiological pathways between these two diseases, and list conventional and non-conventional therapies for patients that suffer from both pathologies. While the topic is not new and others have already studied the connections between NAFLD and psoriasis, the authors have included some recent studies on the topic, and I think that researchers in the field would benefit from the publication of this manuscript.
However, I would recommend that the authors slightly adjust the text according to the minor comments below:
- Line 140: colon should be removed as the multiple differences are not all listed afterwards but rather stated within multiple sentences.
- Line 158: please change the subject from “it” to “they” when referring to multiple scores.
- Line 216 & 277: “MetS” is not a diet, but a syndrome. Please introduce the acronym in the text and avoid referring to it as a diet.
- Paragraph 4: the section that refers to all therapeutic options would be easier to follow if the different treatments are listed as numbered or bullet points (possibly listing conventional and non-conventional therapies in two different sections). Alternatively, a table would help the reader to have an overview of the possible treatments at a glance.
Author Response
Dear reviewer,
Thank you so much for your kind remarks and for the work that you’ve done in helping us to get better work.
However, I would recommend that the authors slightly adjust the text according to the minor comments below:
- Line 140: colon should be removed as the multiple differences are not all listed afterwards but rather stated within multiple sentences.
Done accordingly
- Line 158: please change the subject from “it” to “they” when referring to multiple scores.
Done accordingly
- Line 216 & 277: “MetS” is not a diet, but a syndrome. Please introduce the acronym in the text and avoid referring to it as a diet.
Done accordingly
- Paragraph 4: the section that refers to all therapeutic options would be easier to follow if the different treatments are listed as numbered or bullet points (possibly listing conventional and non-conventional therapies in two different sections). Alternatively, a table would help the reader to have an overview of the possible treatments at a glance.
Following your suggestion, we added a table summarising all the discussed treatment options.
Reviewer 2 Report
Comments and Suggestions for Authors
Dear Authors,
Thank you for submitting your paper to IJMS. Below I present some point-by-point comments and suggestions regarding your paper:
- The abstract is too long. Yours is more than 300 words while according to the Author Guidelines for this journal, the maximum number of words is 200 for an abstract. Please correct it accordingly by shortening the abstract
- In the keywords, you can also add: non-alcoholic fatty liver disease
- Regarding the introduction, it would be beneficial to add a figure with the mechanisms describing the relationship between the NAFLD and psoriasis. You can add it here or in paragraph 3
- Could you improve the quality of figure 1? The arrows seem to be a little bit blurred, similar to the text
- Paragraph 3 could be divided into several shorter subparagraphs. In this form, the paragraph is a little bit long and seems hard to read
- Paragraph ‘therapeutic implications’ should be below the ‘correlation between psoriasis and NAFLD’ paragraph
- It would be beneficial to consider adding a table summarizing the key findings of your narrative review
Kind regards
Comments on the Quality of English LanguageIn the paper, some minor mistakes have been detected. I would recommend checking the grammar before the paper is processed further.
Author Response
Dear reviewer,
Thank you so much for your kind remarks and for the time spent helping us to get a better job.
- The abstract is too long. Yours is more than 300 words while according to the Author Guidelines for this journal, the maximum number of words is 200 for an abstract. Please correct it accordingly by shortening the abstract
Done accordingly (190 words).
- In the keywords, you can also add: non-alcoholic fatty liver disease
Done accordingly
- Regarding the introduction, it would be beneficial to add a figure with the mechanisms describing the relationship between the NAFLD and psoriasis. You can add it here or in paragraph 3
- Could you improve the quality of figure 1? The arrows seem to be a little bit blurred, similar to the text
We significantly improved the figure, redesigning it completely, to meet both of your suggestions.
- Paragraph 3 could be divided into several shorter subparagraphs. In this form, the paragraph is a little bit long and seems hard to read
Done accordingly
- Paragraph ‘therapeutic implications’ should be below the ‘correlation between psoriasis and NAFLD’ paragraph
Actually, we did not modify it, and please allow us to explain why. We consider that the most important part of the two you mention here is the one concerning the correlations between NAFLD and psoriasis, and we believe that it would be more highlighted if in the end, focusing attention on it.
- It would be beneficial to consider adding a table summarizing the key findings of your narrative review
Done accordingly